# Translational Pigeonpea Genomics Consortium for Accelerating Genetic Gains in Pigeonpea (*Cajanus cajan* L.)

**Rachit K. Saxena** [1,*] , **Anil Hake** [1] , **Anupama J. Hingane** [1] , **C. V. Sameer Kumar** [2] ,
**Abhishek Bohra** [3] , **Muniswamy Sonnappa** [4] , **Abhishek Rathore** [1] , **Anil V. Kumar** [1] ,
**Anil Mishra** [5] , **A. N. Tikle** [5] , **Chourat Sudhakar** [2] , **S. Rajamani** [6] , **D. K. Patil** [7] , **I. P. Singh** [3] ,
**N. P. Singh** [3] **and Rajeev K. Varshney** [1,*]

[1]  International Crops Research Institute for the Semi-Arid Tropics (ICRISAT),
   Patancheru 502324, Telangana State, India; H.Anil@cgiar.org (A.H.); H.Anupama@cgiar.org (A.J.H.);
   a.rathore@cgiar.org (A.R.); anil.kumar@cgiar.org (A.V.K.)
[2]  Professor Jayashankar Telangana State Agricultural University (PJTSAU), Rajendranagar,
   Hyderabad 500030, Telangana State, India; sameerkumar1968@gmail.com (C.V.S.K.);
   chouratsudhakar2@gmail.com (C.S.)
[3]  ICAR—Indian Institute of Pulses Research (IIPR), Kanpur 208024, Uttar Pradesh, India;
   abhi.omics@gmail.com (A.B.); ipsingh1963@yahoo.com (I.P.S.); npsingh.iipr@gmail.com (N.P.S.)
[4]  Agricultural Research Station (ARS)—Kalaburagi, University of Agricultural Sciences (UAS),
   Raichur 584104, Karnataka, India; muniswamygpb@gmail.com
[5]  Rajmata Vijayaraje Scindia Krishi Vishwa Vidyalaya (RVSKVV), RAK College of Agriculture (RAKCA),
   Sehore 466001, Madhya Pradesh, India; anil1961.mishra@gmail.com (A.M.); antiklepb@gmail.com (A.N.T.)
[6]  Regional Agricultural Research Station, Lam Farm, Guntur 5220324, Andhra Pradesh, India;
   srajamanibreeder@gmail.com
[7]  Agricultural Research Station (ARS)—Badnapur, Vasantrao Naik Marathwada Krishi Vidyapeeth (VNMKV),
   Parbhani 431402, Maharashtra, India; dkpatil.05@gmail.com
\*  Correspondence: r.saxena@cgiar.org (R.K.S.); r.k.varshney@cgiar.org (R.K.V.)

**Abstract:** Pigeonpea is one of the important pulse crops grown in many states of India and plays a major role in sustainable food and nutritional security for the smallholder farmers. In order to overcome the productivity barrier the Translational Pigeonpea Genomics Consortium (TPGC) was established, representing research institutes from six different states (Andhra Pradesh, Karnataka, Madhya Pradesh, Maharashtra, Telangana, and Uttar Pradesh) of India. To enhance pigeonpea productivity and production the team has been engaged in deploying modern genomics approaches in breeding and popularizing modern varieties in farmers' fields. For instance, new genetic stock has been developed for trait mapping and molecular breeding initiated for enhancing resistance to fusarium wilt and sterility mosaic disease in 11 mega varieties of pigeonpea. In parallel, genomic segments associated with cleistogamous flower, shriveled seed, pods per plant, seeds per pod, 100 seed weight, and seed protein content have been identified. Furthermore, 100 improved lines were evaluated for yield and desirable traits in multi-location trials in different states. Furthermore, a total of 303 farmers' participatory varietal selection (FPVS) trials have been conducted in 129 villages from 15 districts of six states with 16 released varieties/hybrids. Additionally, one line (GRG 152 or Bheema) from multi-location trials has been identified by the All India Coordinated Research Project on Pigeonpea (AICRP-Pigeonpea) and released for cultivation by the Central Variety Release Committee (CVRC). In summary, the collaborative efforts of several research groups through TPGC is accelerating genetics gains in breeding plots and is expected to deliver them to pigeonpea farmers to enhance their income and improve livelihood.

**Keywords:** pigeonpea; genomics; TPGC; FPVS; multi-location trials

## 1. Introduction

Pigeonpea is a pulse crop grown in many countries of the world and plays an important role in sustainable nutritional food security. India ranks first in pigeonpea cultivation area (5.58 mha) and production (4.29 mt) in the world [1]. In the last five years, productivity of pigeonpea in India has shown an increasing trend (11.42%) from 693 (2009–2013) to 774 kg/ha (2014–2018), however, it is lower by ~10% compared to world productivity (761 kg/ha in 2009–2013 and 850 kg/ha in 2014–2018) [1]. Moreover, disproportionate yield gaps between research plots and in farmers' fields of a given variety are also a major concern in India [2]. On the other hand, demand for the pulses is continuously increasing and it has been estimated that 32 million tons of pulses will be required by the year 2030 and 50 million tons of pulses by year 2050 (Vision 2050: Indian Institute of Pulses Research, 2013, www.iipr.res.in). To match these requirements, pulse breeders have been engaged in developing superior varieties but could not achieve the daunting task. In recent times molecular breeding approaches have been successful in developing superior varieties and enhance the production of cereal crops, like rice [3–6], wheat [7–11], sorghum [12–14], maize [15–17], and pearl millet [18,19], and also in legume crops, such as chickpea [20–24] and soybean [25–27]. However, such approaches have not been used until recently in pigeonpea, primarily due to limited information on genes/markers associated with traits. In this direction, the International Initiative on Pigeonpea Genomics (IIPG) decoded and published the genome sequence of pigeonpea in 2012 [28]. As a result of this breakthrough, a significant amount of genomic information has become available [29,30]. However, the availability of the genome sequence or the large-scale of molecular markers alone was not enough to improve crop productivity. These resources can be used as tools to harness the genetic diversity present in the germplasm collection for enhancing the precision and efficiency of crop improvement programs. Therefore, just after the decoding of the pigeonpea genome, a series of consultations with a large number of stakeholders, including the Department of Agriculture Cooperation and Farmers Welfare (DACFW), Indian Council of Agricultural Research (ICAR), several state agricultural universities (SAUs), the private sector, and USAID-India, were conducted to use genome sequence information for translational genomics research for crop improvement. As a continuous effort in translational genomics research for pigeonpea improvement, with funding support from DACFW, the Translational Pigeonpea Genomics Consortium (TPGC) of nine research institutions/agricultural universities representing six different states of India was established in 2017 (Figure 1). During the past three years TPGC has made significant progress (described in the sections below) including: (a) development of new genetic stock for trait mapping, (b) deployment of genomics-assisted breeding in 11 popular varieties of pigeonpea, (c) evaluation of 100 improved lines for their performance in multi-location trials, and (d) demonstration of improved crop varieties in more than 303 farmers' fields across 129 villages from 15 districts of six states (Andhra Pradesh, Karnataka, Madhya Pradesh, Maharashtra, Telangana, and Uttar Pradesh). Several improved lines have also been put in the varietal identification pipeline of Indian Council of Agricultural Research. Molecular markers associated with seed protein content, diseases resistance and yield contributing traits, improved lines with higher yield potential and disease resistance were also identified. In summary, the TPGC has been established with an aim to deploy modern genomics information for pigeonpea improvement, develop/identify new and improved varieties, and to enhance the adoption of superior lines in farmers' fields. The present article reports the significant research achievements of the TPGC as international public goods (IPGs) that will be helping and guiding future pigeonpea improvement programs. Furthermore, this article may also inspire other less-studied crop communities to take similar consortium-based approaches for crop improvement.

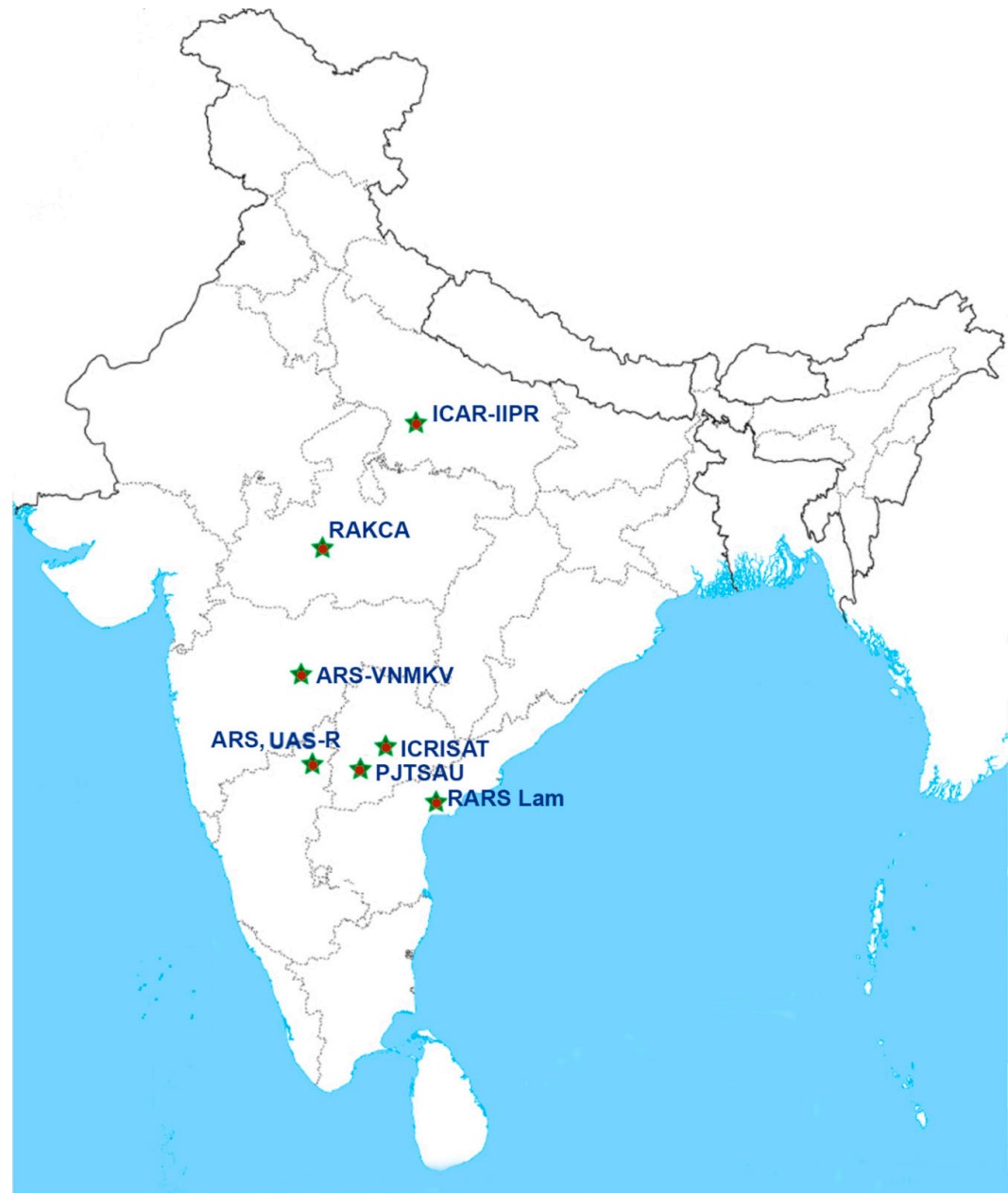

**Figure 1.** Translational Pigeonpea Genomics Consortium: Marked places on the map shows indicative locations of experimentation. ICAR-IIPR: ICAR- Indian Institute of Pulses Research (IIPR), Kanpur, Uttar Pradesh; RAKCA: RAK College of Agriculture (RAKCA), Sehore, Madhya Pradesh; ARS-VNMKV: Agricultural Research Station (ARS)-Badnapur, Vasantrao Naik Marathwada Krishi Vidyapeeth (VNMKV), Parbhani, Maharashtra; ARS, UAS-R: Agricultural Research Station (ARS)-Kalaburagi, University of Agricultural Sciences (UAS), Raichur, Karnataka; ICRISAT: International Crops Research Institute for the Semi-Arid Tropics (ICRISAT), Patancheru, Telangana; PJTSAU: Professor Jayashankar Telangana State Agricultural University (PJTSAU), Telangana; RARS Lam: Regional Agricultural Research Station, Lam Farm, Guntur, Andhra Pradesh.

## 2. Advances in Genetics and Genomics

### 2.1. Novel Breeding and Genetic Materials

A range of genetic and breeding material has been developed in pigeonpea during last few years for their effective use in genomics [30]. This material includes segregating bi-parental populations for target

traits [31–35], diverse genetic stocks including reference set [36], core and mini-core [37,38], and back-cross populations [39]. Further, to reap the advantages of family based mapping, the development of multi-parent mapping populations was initiated in the year 2012. During the last two years these family-based mapping populations, especially Nested Association Mapping Population (NAM), have reached the recombinant inbred line (RIL) stage. At present, multi-location evaluation of the NAM population is underway. Another family-based mapping population i.e., Multi-Parent Advanced Generation Inter-Cross (MAGIC), has also been advanced during the last two years and it has reached to RIL stage in the year 2019–2020. The generation of these family-based mapping populations in pigeonpea have provided new genetic combinations for trait discovery and also for genomics applications for new cultivar development. The significant features of these NAM and MAGIC populations have been provided below.

*2.2. Nested Association Mapping Population (NAM)*

The NAM population consisting of 2224 RILs in pigeonpea was developed by crossing 10 pigeonpea diverse founder lines as female parent to a common pollen parent ICPL 87119 (Table 1, Figure 2). The seeds of 2224 RILs of the NAM population were sown in an augmented block design with spacing of 45 × 30 cm in a single row of 1 m length during the 2018–2019 cropping season. The first set of phenotyping data on the stabilized NAM population ($F_6$ plants) was collected in cropping season 2018–2019 at ICRISAT. The year 2018–2019 was also used for seed multiplication of the NAM population so that planned multi-location trials could be conducted. During the year 2018–2019, RILs of the NAM population were evaluated for agronomic traits including days to first flower, days to 50% flowering, days to 75% maturity, number of primary and secondary branches, pods per plant, pod and grain weight per plant, and 100 seed weight as per the pigeonpea descriptor [40]. Preliminary analysis of phenotyping data collected on NAM population showed significant variations for the above mentioned traits in RILs. For instance, a range of 60 days to 141 days with a mean value of 95.14 days has been observed for days to 50% flowering across NAM population (Figure 3). Similarly, RILs in the NAM population matured in 110–218 days with a mean value of 146.91 days. In the case of two important yield measuring/contributing traits, i.e., seeds per pod (1.2–6.8) and 100-seed weight (3.37–16.99 g), a wide range of variation has been observed. It is important to mention that multi-location evaluation of NAM over two or more locations per family in six states for two years has been planned. The comprehensive data analysis after multi-location evaluation will provide exact information on the available phenotypic variability in the NAM population and to identify elite lines suitable for major agroecologies of pigeonpea cultivation in India.

**Table 1.** Characteristic features of parents used in the development of Nested Association Mapping (NAM) population in pigeonpea.

| Genotypes | Features |
|---|---|
| **Nested Parent** | |
| **Asha (ICPL 87119)** | Genome sequence available, leading variety, resistant to FW and SMD |
| **Founder Parents** | |
| **HPL 24** | High protein content, medium duration, compact, susceptible to FW and resistant to SMD, inter-specific derivative |
| **ICP 7035** | Medium duration, SMD resistant, large purple seed, high sugar |
| **ICP 8863** | Mid-late, highly resistant to FW and susceptible to SMD, an extensively grown variety in Northern Karnataka and Maharashtra region of India |

**Table 1.** *Cont.*

| Genotypes | Features |
|---|---|
| **ICPL 87** | Early duration, determinate, short, high combiner |
| **ICPL 88039** | Extra early maturity, indeterminate, good yield |
| **ICPL 85063** | Medium duration, indeterminate, good yield, more branching |
| **MN 1** | Super early, small seeded, determinate |
| **ICP 28** | Early maturity, local variety |
| **ICPL 85010** | Early maturity, local variety |
| **ICP 7263** | Determinate, long podded, white seeded |

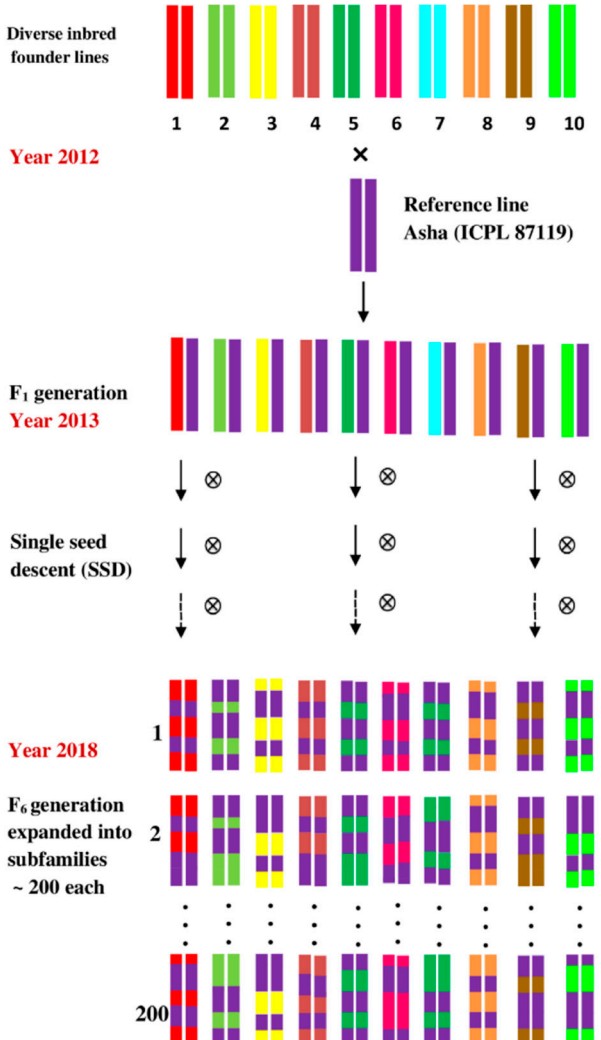

**Figure 2.** Nested association mapping (NAM) population in pigeonpea. Inbred, Asha (ICPL 87119) was used for crossing with 10 diverse inbred lines separately. The hybrids from each of 10 crosses were selfed to develop $F_2$s. From these $F_2$ seeds, at least 200 progenies were generated in each of 10 crosses by single seed decent method to take these lines to $F_6$ generation.

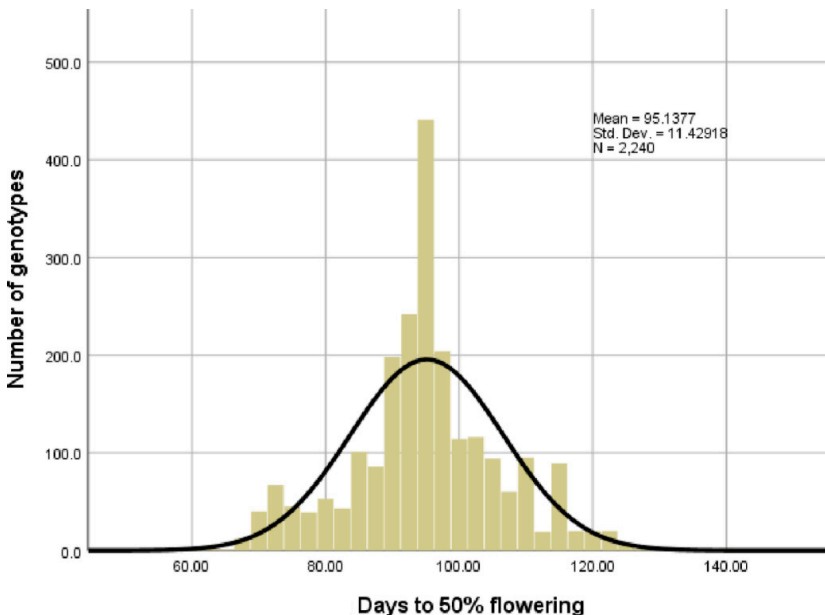

**Figure 3.** Frequency distribution for days to 50% flowering in NAM population of pigeonpea. A range of 60 days to 141 days with a mean value of 95.14 days has been observed for days to 50% flowering across NAM population.

## 2.3. Multi-Parent Advanced Generation Inter-Cross (MAGIC)

To bring diversity from landraces and superior varieties, the MAGIC population has been developed as per the crossing scheme of Cavanagh et al. [41] using eight crossing parents (ICP 7426, HPL 24, ICP 11605, ICP 14209, ICP 14486, ICP 5529, ICP 7035, and ICP 8863) by 28 two-ways, 14 four-ways, and seven eight-ways crosses (Table 2, Figure 4). These eight parents have significant variations for agronomic (pod numbers per plant, maturity, days to flowering, grain yield), quality (sugar and protein content), and disease resistance (fusarium wilt and sterility mosaic disease) traits. The $F_6$ seeds for this population were harvested from the $F_5$ plants in year 2018–2019 and seed multiplication was undertaken for $F_6$ plants in cropping season 2019–2020. Homozygous lines (~1300 RILs) obtained from multi-parent crossing approach will be used for high resolution trait mapping that is otherwise not possible by using conventional bi-parental mapping populations. This population also offers new breeding material with enhanced diversity and combined desirable traits (e.g., early maturity, high seed protein content, higher yield, and disease resistance). Some of these lines can be used as parents in future breeding programs or can be put in the varietal release pipelines.

**Table 2.** Characteristic features of parents used in the development of Multi-Parent Advanced Generation Inter-Cross (MAGIC) population of pigeonpea.

| Genotypes | Features |
| --- | --- |
| ICP 7426 | High pod numbers, medium duration |
| HPL 24 | High protein content, medium duration, compact, susceptible to FW and resistant to SMD, inter-specific derivative |
| ICP 11605 | Early flowering, germplasm line |
| ICP 14209 | High number of pods, germplasm line |
| ICP 14486 | Early flowering, germplasm line |
| ICP 5529 | Medium duration, obcordate leaves, compact plant, modified flower |
| ICP 7035 | Medium duration, SMD resistant, large purple seed, high sugar content |
| ICP 8863 | Mid-late, highly resistant to FW and susceptible to SMD, an extensively grown variety in Northern Karnataka and Maharashtra region of India |

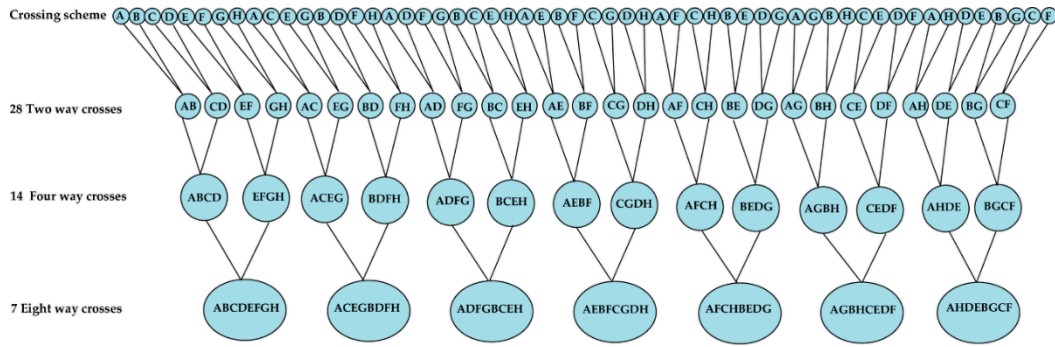

**Figure 4.** Development of a MAGIC population in pigeonpea. Eight genotypes (ICP 7426, HPL 24, ICP 11605, ICP 14209, ICP 14486, ICP 5529, ICP 7035 and ICP 8863) were used in crossing program as parents to develop MAGIC population. Crossing scheme included, 28 two-ways, 14 four-ways and 7 eight-ways crosses. Further, hybrids coming from 7 eight-way crosses were selfed to generate ~1300 $F_6$ lines.

## 3. Genomics Advances and Genomics-Assisted Breeding

### 3.1. Marker Assisted Back-Crossing for Fusarium Wilt (FW) and Sterility Mosaic Diseases (SMD) Resistance

Fusarium wilt (FW) and sterility mosaic diseases (SMD) are considered as major yield reducing biotic stresses in pigeonpea [42–45]. Pathogen variability have further added to the severity of these diseases [46–48]. Moreover, the known resistant varieties are witnessing the breakdown of resistance. Therefore, it has been planned to introduce, combine, or reconstruct the resistance for FW and SMD in leading pigeonpea varieties from different agro-climatic zones of India following marker-assisted back-crossing (MABC). In this endeavor, backcross populations were developed by crossing 11 mega varieties (recurrent parent, susceptible to FW and/or SMD) with ICPL 20096 (donor parent, resistant to FW and SMD) following two cycles of backcrosses (Table 3) during cropping seasons (rainy) 2017–2018 to 2018–2019. These 11 released varieties were crossed with ICPL 20096 as recipient parents to generate $F_1$s in respective crossing combinations. True $F_1$s from respective crosses were identified using molecular markers. Subsequently confirmed $F_1$s of respective crosses were used to make backcrosses with the recipient or recurrent parent. The backcross seeds ($BC_1F_1$) from respective crosses were harvested and tested with markers for foreground selection and second backcrossing (Table 4). It is important to note that we have developed two different sets of 10 markers each associated with FW and SMD resistance (unpublished). These markers have been used for foreground selection in $BC_1F_1$ plants. Those $BC_1F_1$ plants carrying the highest heterozygosity for 10-marker panel were used for second round of backcrossing with respective recurrent parents for maximum genome recovery. In this way, at present, we have reached the stage of $BC_2F_1$ seeds. It has been planned to advance and obtain $BC_2F_2$ seeds from $BC_2F_1$ plants in upcoming years. Further selfed $BC_2F_2$ seeds from each cross will be sown in sick plot for evaluation of FW and SMD incidences. Most promising lines identified for FW and SMD resistance will be subjected to yield evaluations in hotspot regions of the country. The MABC-bred improved lines showing higher disease resistance and similar or better yield performance as compared to recurrent parent in both stressed and normal conditions will enter varietal identification and release pipelines.

**Table 3.** List of pigeonpea mega varieties targeted for introgression of resistance to fusarium wilt (FW) and sterility mosaic disease (SMD) resistance.

| Varieties | # Centre | Introgression of the Trait |
|---|---|---|
| LRG 41 | RARS-Lam | FW and SMD |
| LRG 52 | RARS-Lam | FW and SMD |
| ICPL 88039 | ICRISAT | FW |
| UPAS 120 | ICRISAT | FW |
| ICP 8863 | ARS-Kalaburagi | SMD |
| TS 3R | ARS-Kalaburagi | SMD |
| TJT 501 | RAKCA-Sehore | FW |
| JKM 189 | RAKCA-Sehore | FW |
| BDN 711 | ARS-Badnapur | FW |
| PRG 176 | PJTSAU | FW and SMD |
| Bahar | ICAR-IIPR | FW |

# Centre undertaking marker-assisted backcrossing program in selected preferred varieties in the region.

**Table 4.** Details and current status of marker assisted backcrossing for improving elite lines for resistance to fusarium wilt and sterility mosaic disease.

| S. No. | Recurrent Parent | Donor Parent | $F_1$ Plants | $F_1s$ Confirmed | No. of $BC_1F_{1s}$ Grown | No. of $BC_1F_{1s}$ Confirmed |
|---|---|---|---|---|---|---|
| 1 | BDN 711 | ICPL 20096 | 17 | 17 | 92 | 85 |
| 2 | ICP 8863 | ICPL 20096 | 30 | 21 | 92 | 78 |
| 3 | TS 3R | ICPL 20096 | 26 | 22 | 92 | 80 |
| 4 | TJT 501 | ICPL 20096 | 28 | 27 | 19 | 18 |
| 5 | JKM 189 | ICPL 20096 | 25 | 7 | 60 | 52 |
| 6 | Bahar | ICPL 20096 | 30 | 28 | 92 | 89 |
| 7 | PRG 176 | ICPL 20096 | 11 | 9 | 5 | 4 |
| 8 | LRG 41 | ICPL 20096 | 11 | 11 | 5 | 5 |
| 9 | LRG 52 | ICPL 20096 | 4 | 2 | 7 | 6 |
| 10 | ICPL 88039 | ICPL 20096 | 11 | 11 | 132 | 122 |
| 11 | UPAS 120 | ICPL 20096 | 28 | 15 | 88 | 65 |

*3.2. Development of Trait-Associated Markers*

Availability of molecular markers governing various traits are prerequisites for deploying genomics-assisted breeding in crop improvement. While the MABC approach is being deployed for improving resistance to FW and SMD, molecular mapping for several quality- and yield-related traits was also undertaken by TPGC. These findings have been considered as important milestones for pigeonpea improvement. A summary on the development of molecular markers has been presented below:

A bi-parental trait mapping study by Yadav et al. [49] identified 26 quantitative trait loci (QTLs) on nine CcLGs (except CcLG01 and CcLG05) for 10 traits viz. cleistogamous flower (Cl), shriveled seed (ShS), plant height (PH), number of primary branches (PB), number of secondary branches (SB), pods per plant (PP), seeds per pod (SP), seed count per plant (SC), seed weight (SW), and seed size (100 seed weight) (SS) using RILs of ICPL 99010 × ICP 5529. The phenotypic variance explained (PVE) by each QTL ranged from 9.1% to 50.6% (Table 5, Table S1). Similarly, a total of 17 CAPS/dCAPS markers were identified for seed protein content (SPC) using high (HPL 24, ICP 5529) and low (ICP 11605, ICPL 87119) seed protein content (SPC) containing pigeonpea genotypes by whole genome re-sequencing (WGRS) data [50]. Further, these markers were validated using $F_2$ population of ICP 5529 × ICP 11605. Out of 17 CAPS/dCAPS markers, four genic-CAPS/dCAPS markers, spc003 (NADH-GOGAT), spc107 (copper transporter), spc017 (protein kinase), and spc100 (BLISTER) revealed co-segregation with SPC (Table S1). Using five $F_2$ populations, Obala et al. [51] identified 192 QTLs across 10 CcLGs for five traits, namely, seed protein content (SPC) seed weight (SW), seed yield (SY), growth habit (GH),

and days to first flowering (DFF) with PVE of 0.7–91.3%. Major effect (PVE $\geq$ 10%) QTLs included 14 QTLs for SPC, 16 QTLs for SW, 17 QTLs for SY, 19 QTLs for GH, and 24 QTLs for DFF (Table S1).

**Table 5.** Summary on QTLs/genomic segments identified for target traits in pigeonpea.

| Trait | Number of QTLs/Genomic Segments Identified | # PVE Range (%) | Reference |
|---|---|---|---|
| Cleistogamy | 5 | 9.10–50.60 | Yadav et al. 2019 |
| Seed shape | 3 | 11.80–37.20 | Yadav et al. 2019 |
| Seed size | 2 | 29.50–33.90 | Yadav et al. 2019 |
| Seed protein content | 19 | 2.20–23.50 | Obala et al. 2019; 2020 |
| 100 seed weight | 18 | 10.10–46.60 | Obala et al. 2020 |
| Seed yield | 18 | 10.20–53.00 | Obala et al. 2020 |
| Growth habit | 21 | 10.90–91.30 | Obala et al. 2020 |
| Days to first flowering | 28 | 10.90–47.60 | Obala et al. 2020 |

[#] Phenotypic variation explained.

## 4. Promising Pigeonpea Lines Identified through Multi-Location Trials

To select high yielding superior lines from multi-location trials, three trials (one each for super-early (>90 days), early (140–160 days), and medium maturity (>160–180 days) group) were conducted during cropping seasons of 2017–2018 (year 1) and 2018–2019 (year 2) for grain yield (Figure 5, Table S2). Thirty genotypes of super-early maturity group were tested in year 1 at five centers, namely, Kalaburagi, Kanpur, Tandur, Lam, and Patancheru, while in year 2, it was tested at three centers, namely, Tandur, Lam, and Kanpur. Likewise, 30 genotypes of the early maturity group were tested in year 1 at Kanpur, Tandur, Lam, Kalaburagi, and Patancheru, while in year 2, these were tested at Kanpur, Tandur, Lam, and Badnapur. The 40 genotypes of the medium-maturity group were tested at Kalaburagi, Tandur, and Lam for two consecutive years, while one time these were tested at Patancheru (year 1), Badnapur (year 2), and Sehore (year 2). In multi-location trials, seeds of each entry were sown in four rows of 3 m length with spacing of 90 × 30 cm, 45 × 20 cm, and 30 × 10 cm for medium/late duration, early and super-early maturity groups, respectively, in a randomized complete block design with three replications. The mean for each genotype in a replication was calculated using the observations recorded from the whole plot. The overall mean for each genotype was calculated using the values from each replication. Minimum and maximum means of the genotypes were considered to record the range. Individual analysis of variance (ANOVA) was carried out in order to partition the variation due to different sources following the method of Panse and Sukhatme [52]. Combined ANOVA was computed using a general linear mixed model using the procglm function of SAS version 9.2 [53]. The stability analysis of selected genotypes for grain yield was done using the data recorded during the rainy season of 2017–2018 and 2018–2019 across various locations. A GGE biplot (site regression analysis) was used to illustrate the genotype plus genotype-by-environment variation using principal component (PC) scores from singular value decomposition (SVD). A GGE biplot with average-environment coordination (AEC) and polygon view was drawn to examine the performance of all genotypes within a specific environment and to simultaneously select genotypes based on stability and mean performance [54]. The model for the GGE based on SVD of the first two PCs is given by:

$$Y_{ij} - \mu - \beta_j = \lambda_1 \xi_{i1} \eta_{j1} + \lambda_2 \xi_{i2} \eta_{j2} + \varepsilon_{ij} Y_{ij} - \mu - \beta = \lambda_1 \xi_{i1} \eta_{j1} + \lambda_2 \xi_{i2} \eta_{j2} + \varepsilon_{ij}$$

where $Y_{ij}$ is the mean performance of genotype i in environment j, $\mu$ is the grand mean, $\beta_j$ is the environment j main effect, $\lambda_1$ and $\lambda_2$ are the singular values of the first and second PC, $\xi_{i1}$ and $\xi_{i2}$ are the eigenvectors for genotype i, $\eta_{j1}$ and $\eta_{j2}$ are the eigenvectors for environment j, and $\varepsilon_{ij}$ is the residual effect.

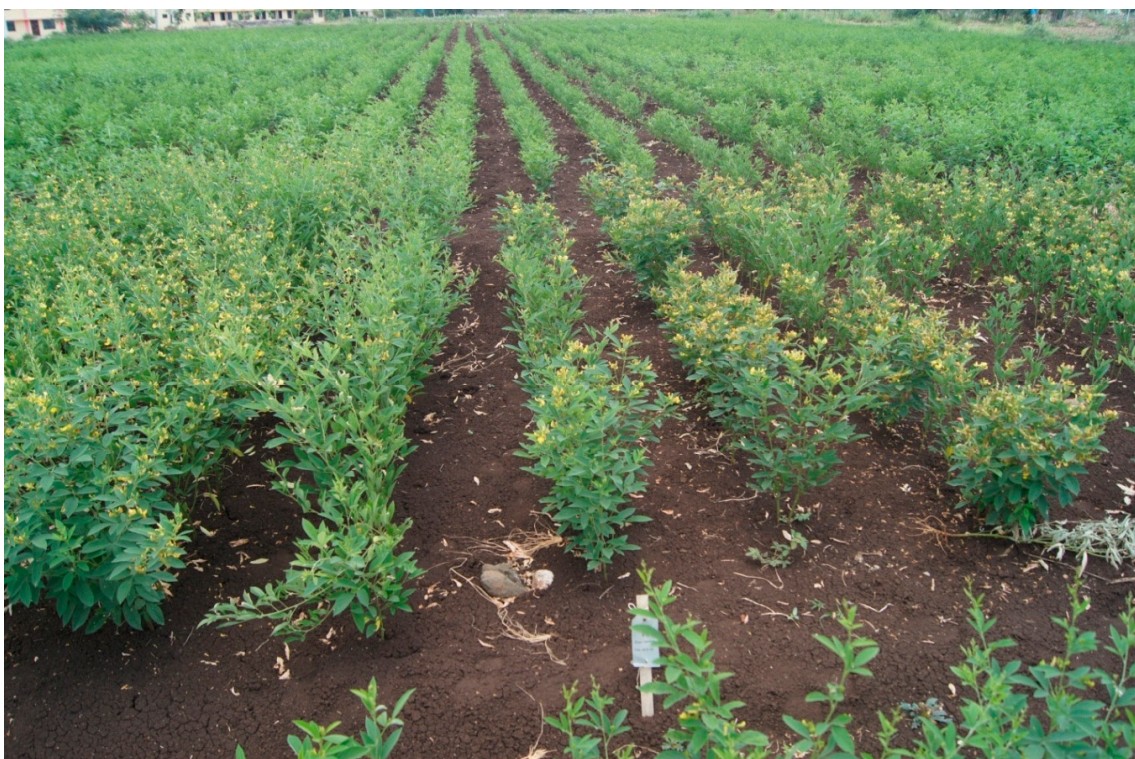

**Figure 5.** An overview of pigeonpea field with super-early pigeonpea lines at flowering stage and remaining lines at vegetative stage.

### 4.1. Performance and Stability of Genotype and Environment for Grain Yield

In super-early, early- and medium-maturity groups of multi-location trials, individual (Table S3) as well as combined analysis of variance (ANOVA) (Table 6) revealed significant differences in genotype, environments, and genotype × environment interaction (GEI) effects for grain yield. To know the best test environment and superior genotype (high yield and stable), GGE ((genotype (G) + (genotype (G) × environment (E) interaction)) biplot analysis was conducted with the phenotyping data recorded in multi-location trials. A GGE biplot explained 73.58, 66.53, and 67.85% of total variation of the environment-centered G by E table for grain yield for super-early, early and medium duration trials respectively. GGE biplot analysis revealed two mega environments in each trial. For instance, in super-early, Tandur and rest environments (Kalaburagi_2017, Patancheru_2017, Kanpur and Lam), for early, Patancheru and rest environments (Kanpur, Tandur, Lam, Kalaburagi and Badnapur) and for medium duration trial, Badnapur, Lam_2018 and rest environments (Tandur, Kalaburagi, Patancheru and Lam_2017) were observed for the performance of grain yield (Figure 6). In super-early trials, Lam_2017 and Tandur (2017 and 2018) identified as more discriminating environment while Kalaburagi, Patancheru, Kanpur, and Lam_2018 identified as least discriminating environment for grain yield (Figure 6a). Likewise, in early duration trial, Lam_2017 was identified as the most discriminating, however, Kalaburagi_2017, Kanpur_2017, and Tandur_2017 were identified as average discriminating, while Kanpur_2018, Lam_2018, Badnapur_2017, and Patancheru_2017 were identified as the least discriminating environments for grain yield (Figure 6b). In medium-duration trials, Sehore_2018 is the most discriminating, Tandur is average discriminating, while rest environments Lam, Kalaburagi, Patancheru, and Badnapur were identified as least discriminating environments for grain yield (Figure 6c).

**Table 6.** Combined analysis of variance for grain yield of super-early, early and medium duration trials during cropping season 2017–2018 and 2018–2019.

| Trial | Effects | | | | |
|---|---|---|---|---|---|
| | | Environment | Rep (Env) | Genotype | Genotype × Environment |
| Super-early (30 genotypes) | df | 7.0 | 16.0 | 28.0 | 196.0 |
| | F | 2245.8 ** | 10.7 ** | 14.2 ** | 17.9 ** |
| Early (30 genotypes) | df | 7.0 | 14.0 | 29.0 | 203.0 |
| | F | 1305.0 * | 7.9 ** | 24.8 ** | 15.6 ** |
| Medium (40 genotypes) | df | 8.0 | 15.0 | 34.0 | 272.0 |
| | F | 1093.2 ** | 11.7 ** | 22.0 ** | 14.1 ** |

*: Significant at 0.05 probability level, **: Significant at 0.01 probability level.

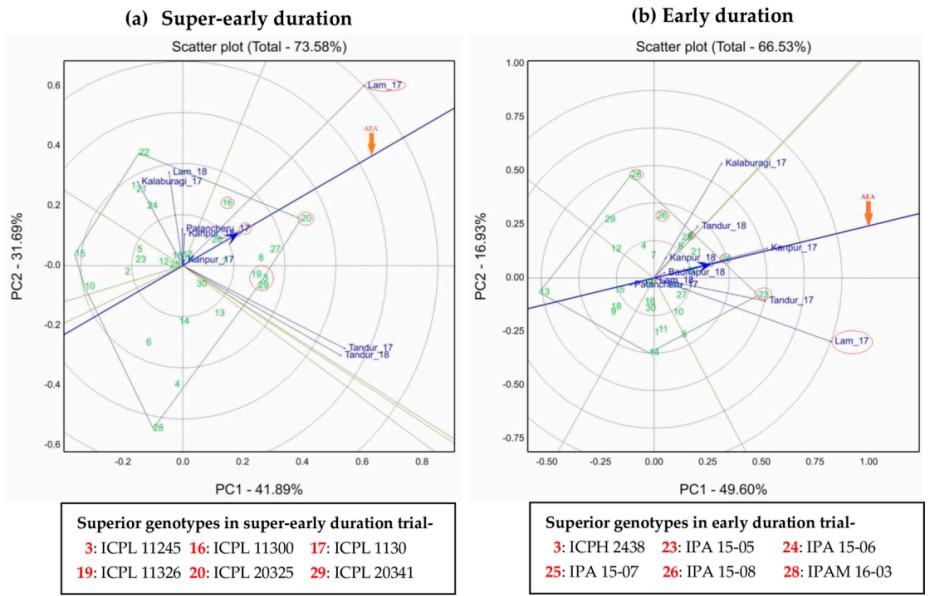

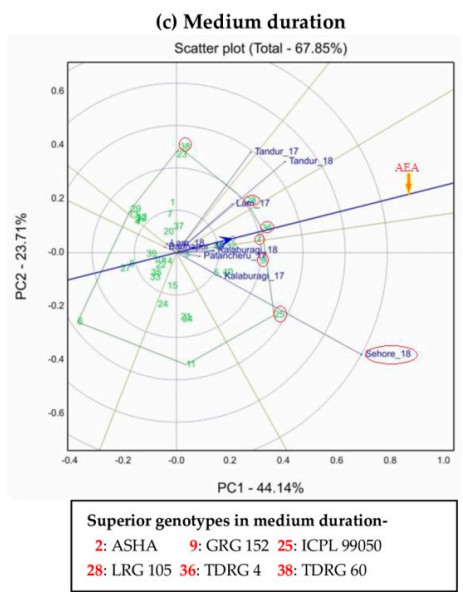

**Figure 6.** GGE biplot showing ranking of genotypes and environments for mean and stability for grain yield in (**a**) super-early, (**b**) early and (**c**) medium duration trials over locations in India during cropping season 2017–2018 and 2018–2019.

*4.2. Promising Lines in Super-Early Duration Trial*

Pigeonpea lines, namely, ICPL 20325 at Tandur, ICPL 11300 at Lam and ICPL 11279 at Kanpur exhibited yield advantage (14.8–154.4%) over MN1 (check) across the years (Table S4). Whereas, ICPL 20326 at Kalaburagi and ICPL 11301 at Patancheru have also shown yield advantages of 20.2–109% as compared to check (Table S4). One pigeonpea line, ICPL 20325, has been identified as a superior line with higher grain yield and high stability over the years and across the locations (Figure 6a) with 24.09% yield advantage over MN1 (Table S4).

*4.3. Promising Lines in Early Duration Trial*

IPA 15-05 at Lam and Tandur and IPA 15-06 at Kanpur revealed yield advantage (33.3–127.9%) over ICPL 88039 (check) across the years (Table S5). Whereas, on the other testing locations, IPAM 16-03 at Kalaburagi, IPA 15-08 at Patancheru and ICPL 92047 at Badnapur exhibited significant yield advantage (4.3–118.2%) over ICPL 88039 (Table S5). Overall, IPA 15-05 exhibited the highest grain yield and gained a yield advantage of 52.29% over ICPL 88039 across the locations over the years with high stability (Table S5, Figure 6b). Importantly, two lines IPA 15-03 and IPA 15-06 had 7.47 and 8.81% yield advantage, respectively, over the best check when evaluated in initial varietal trial (IVT) of AICRP-Pigeonpea for the northwestern plain zone (NWPZ) and central zone (CZ). Furthermore, with an average yield of 1791 kg/ha, IPA 15-06 has shown 11.44% yield advantage over the best check in CZ in advanced varietal trial 1 (AVT 1) of AICRP-Pigeonpea.

*4.4. Promising Lines in Medium Duration Trial*

ICPL 99050 at Kalaburagi, LRG 105 at Lam, TDRG 60 at Tandur revealed higher grain yield and yield advantage up to 285.9% over the checks across the years (Table S6). At Badnapur, TJT 501 and at Patancheru, AGL 1603-4 revealed a yield advantage (0.6–89.4%) over the checks (Table S6). At Sehore, ICPL 99050 exhibited a yield advantage of 23.7–397.7% over the checks. GRG 152 followed by LRG 105 and ICPL 99050 exhibited higher yield as compared to different checks and stability (Table S6, Figure 6c).

## 5. Enhancing Varietal Adoption through Farmer Participatory Varietal Selection (FPVS) Trials

In order to enhance the adoption of available varieties/hybrids, during the last two years a total of 303 FPVS trials have been conducted in 129 villages of 15 districts of six states, namely, Andhra Pradesh, Karnataka, Madhya Pradesh, Maharashtra, Telangana, and Uttar Pradesh. FPVS trials were conducted using 4–5 improved cultivars for the evaluation of grain yield during rainy season of 2017–2018 (year 1) and 2018–2019 (year 2) (Table 7). The seeds of each entry for FPVS trial were sown in 1000 square meter plots on farmers' fields. Data analysis for the FPVS trials was conducted following similar methods mentioned in multi-location trials.

**Table 7.** List of varieties/hybrids used for farmer participatory varietal selection trial in the six states of India during cropping season 2017–2018 and 2018–2019.

| SN | State | Districts | Variety/Hybrids [#] | No. of FPVS | | Total FPVS |
|---|---|---|---|---|---|---|
| | | | | 2017–2018 | 2018–2019 | |
| 1 | Andhra Pradesh | Guntur, Prakasam, and Kurnool | LRG 52, **LRG 105, LRG 160**, ICPH 2433, and ICPH 2740 | 30 | 30 | 60 |
| 2 | Karnataka | Kalaburagi, Bida, r and Yadgir | TS 3R, GRG 811, ICPL 332, **BSMR 736** and ICPH 2433 | 25 | 30 | 55 |
| 3 | Madhya Pradesh | Sehore, Shajapur, and Rajgarh | **ICPH 2671**, ICPH 2443, JKM 189, TJT 501, and ICPL 88039 | 25 | 30 | 55 |
| 4 | Maharashtra | Aurangabad, Jalna, and Parbhani | **BDN 711,** BDN 716, BSMR 853, and BSMR 736 | - | 28 | 28 |

**Table 7.** *Cont.*

| SN | State | Districts | Variety/Hybrids [#] | No. of FPVS | | Total FPVS |
|----|-------|-----------|-------------------|-------------|--|------------|
| | | | | 2017–2018 | 2018–2019 | |
| 5 | Telangana State | Mahabubnagar, Rangareddy, and Warangal | ICPL 161, ICPL 99050, **ICPL 332 WR**, PRG 176, and RGT 1 | 30 | 30 | 60 |
| 6 | Uttar Pradesh | Kanpur, Banda, and Chitrakoot | ICPH 2740, IPA 203, JKM 189, and ICPL 88039 | 25 | 20 | 45 |

[#] Lines in "bold" identified as best performing lines by farmers in respective state.

### 5.1. Performance and Stability of Genotype and Environment for Grain Yield

The combined analysis of variance (ANOVA) revealed significant differences in genotype but non-significant genotype × environment interaction (GEI) effects for grain yield at locations in Karnataka and Maharashtra states over the years (Table 8), while combined ANOVA revealed significant genotype, environment and GEI effects over the years and locations for grain yield in locations at Telangana, Andhra Pradesh, and Madhya Pradesh (Table 8). Stability analysis for grain yield was conducted using a GGE biplot for FPVS where significant GEI was observed at Madhya Pradesh, Andhra Pradesh, and Telangana. A GGE biplot explained 99.48%, 99.14%, and 98.75% of total variation of the environment-centered G by E table for Madhya Pradesh, Andhra Pradesh, and Telangana, respectively. Due to non-significant GEI, the environment had no effect on the performance of genotypes and the stable performance of genotypes was observed at Karnataka (Table S7, Figure 7a) and Maharashtra (Table S8, Figure 7b). In Madhya Pradesh, Sehore was identified as the most discriminating and representative environment, and significantly differed with Shajapur and Rajgarh for grain yield (Table S9, Figure 8a). In Andhra Pradesh, Guntur region is highly discriminating, more representative, and significantly differed with Prakasam and Kurnool for grain yield (Table S10, Figure 8b). In Telangana, Mahabubnagar was identified as the most discriminating region, and significantly differed with Warangal and Rangareddy for grain yield (Table S11, Figure 8c).

**Table 8.** Combined analysis of variance for grain yield of farmer participatory varietal selection trials during cropping season 2017–2018 and 2018–2019.

| FPVS Trial | | Effects | | |
|------------|--|---------|--|--|
| | | Environment | Genotype | Genotype × Environment |
| Andhra Pradesh | df | 2.0 | 4.0 | 8.0 |
| | F | 145.8 ** | 23.78 ** | 6.6 ** |
| Maharashtra | df | 2.0 | 3.0 | 6.0 |
| | F | 0.09 NS | 63.9 ** | 1.36 NS |
| Madhya Pradesh | df | 2.0 | 4.0 | 8.0 |
| | F | 13.5 ** | 109.9 ** | 5.1 ** |
| Karnataka | df | 2.0 | 3.0 | 6.0 |
| | F | 1.3 NS | 9.8 ** | 1.4 NS |
| Telangana | df | 2.0 | 4.0 | 8.0 |
| | F | 4.11 * | 9.56 ** | 14.58 ** |

NS: Non-significant at 0.05 probability level, *: Significant at 0.05 probability level, **: Significant at 0.01 probability level.

### 5.2. Selection of High-Performing Varieties in Different States

In Karnataka over the years, BSMR 736 revealed significantly higher grain yield at Bidar, Kalaburagi, and Yadgir, respectively, with yield advantage of 8.6–29.9% over TS 3R (check). Across the location and over the years, BSMR 736 gained a 22.3% yield advantage over TS 3R (Table S7). In Maharashtra, BDN 711 exhibited the highest grain yield in Aurangabad, Jalna, and Parbhani districts, over BSMR 736 (check), with yield advantage in the range of 18.7–35.2% (Table S8). Across the locations in

Maharashtra, BDN 711 followed by BDN 716 revealed significantly higher grain yield over BSMR 736 with a yield advantage of 23.3–27.5%. In Madhya Pradesh over the years, ICPH 2671 significantly exhibited the highest grain yield at Rajgarh, Shajapur, and Sehore with a yield advantage in the range of 18.3–35.3% over the check variety, ICPL 88039. In Madhya Pradesh, over the years and across the locations, ICPH 2671 revealed superior performance in terms of higher yield, stability, and gained yield advantage of 34.72% over ICPL 88039 (Table S9). In Andhra Pradesh at Guntur, LRG 105 followed by LRG 160 exhibited significantly higher grain yield over check, LRG 52 with yield advantage in the range of 5.1–5.2%. Likewise, at Prakasam, LRG 105 followed by LRG 160 revealed significantly higher grain yield over LRG 52 with yield advantage in the range of 6.9 to 7.2%. At Kurnool, LRG 52 followed by LRG 160 revealed significantly higher grain yield over LRG 52 with yield advantage in the range of 7.1–7.5%. Overall, in Andhra Pradesh, LRG 160 followed by LRG 105 exhibited the highest grain yield with yield advantage in the range of 0.9–4.2% over LRG 52. A GGE biplot indicated LRG 160 was better performing and highly stable across the locations (Table S10). In Telangana, ICPL 332 WR significantly exhibited the highest grain yield at Mahabubnagar, Warrangal, and Rangareddy over check, TS 3R with yield advantage in the range of 6.5–17.8%. Across the location in Telangana ICPL 332 WR revealed the highest grain yield over TS 3R with yield advantage of 14.6% (Table S11).

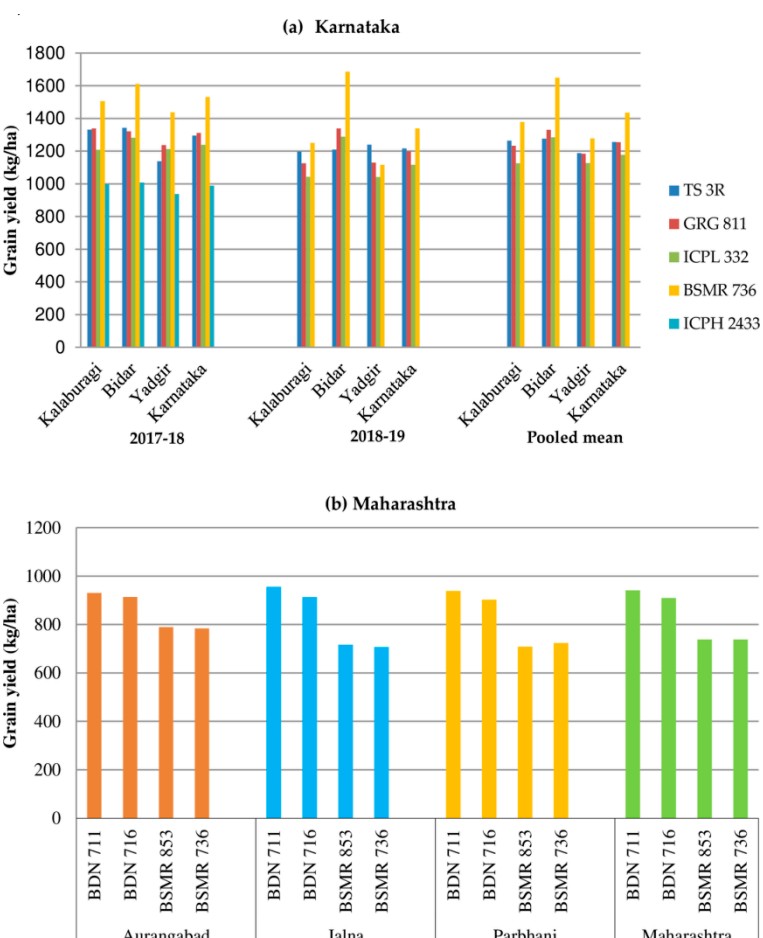

**Figure 7.** Farmers participatory varietal selection trials of pigeonpea in (**a**) Karnataka: BSMR 736 was identified as the best performing line in FPVs trials in Karnataka state. (**b**) Maharashtra: BDN 711 was identified as the best performing line in FPVs trials in Maharashtra state.

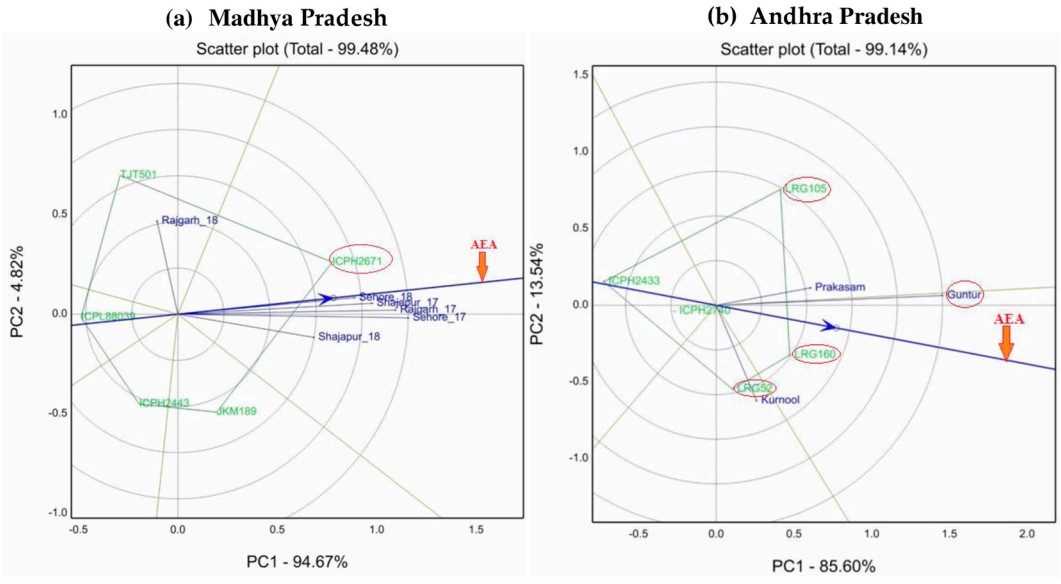

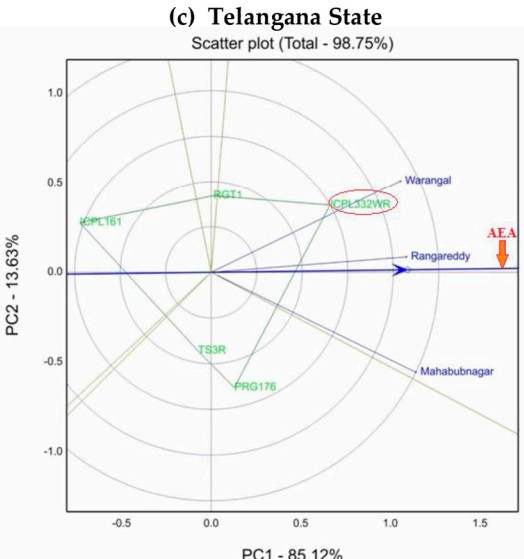

**Figure 8.** GGE biplot showing ranking of genotypes and environments in FPVS trial for mean and stability for grain yield in (**a**) Madhya Pradesh (**b**) Andhra Pradesh and (**c**) Telangana State.

## 6. Summary and Outlook

By using the TPGC as the multi-institutional team, we established a data-driven crop improvement program in pigeonpea. The TPGC has been engaged on different fronts starting from upstream research such as development of traits associated markers to very downstream work such as FPVS trials with farmers. For instance, significant efforts have been made to establish relationships between observed phenotype and genomic constitution in pigeonpea. This will enhance the precision and efficiency of the prediction of phenotypes from genotypes and subsequently in developing superior genotypes and varieties. The TPGC has been successful in developing new genetic stock, trait-associated molecular markers, and is currently working on developing new, promising lines through the MABC approach. Similarly, new genetic stock in the form of MAGIC and NAM populations have been developed. These multi-parent populations not only overcome the limitations of traditional trait mapping but also offer new potential to accurately define the genetic basis of complex crop traits [55]. NAM and MAGIC populations allow intensive genome reshuffling, making them suitable for high-resolution mapping due to broad genetic diversity created through high numbers of recombination events. Genotypes in the

NAM population exhibited a normal distribution for the majority of the traits, indicating quantitative genetic control. The significant variations observed in NAM and MAGIC populations will be harnessed in coming years for identification of QTLs and candidate genes for important traits, like pod numbers, growth habit, protein content, flowering, maturity, seed size, seed colour, etc. As success stories of MABC are available in other legume crops, like chickpea [21] and peanut [56,57], the TPGC has also initiated the introgression QTLs for diseases (FW and SMD) resistance in 11 mega varieties of pigeonpea. We anticipate some improved lines through MABC for commercial release in the near future.

Multi-location trials and FPVS trials were used for identification of high-performing varieties in station plots and farmers' fields, respectively, in different states. In early maturity group multi-location trials, IPA 15-05 revealed the highest grain yield at Tandur, Lam and across the locations while, from medium maturity trials, TDRG 60 followed by TDRG 4 and LRG 105 at Tandur, LRG 105 followed by LRG 52 and JKM 189 at Lam, ICPL 99050 at Kalaburagi and Sehore, and GRG 152 across the locations revealed the highest grain yield over check. Likewise, in super-early, ICPL 20325 followed by ICPL 11245 and ICPL 11292 at Tandur, ICPL 11300 followed by ICPL 20325 and ICPL 20327 at Lam, ICPL 11301 followed by ICPL 11244 and ICPL 20325 at Patancheru and ICPL 20325 across the locations revealed the higher grain yield over check with higher grain yield and high stability. The test environments that are highly discriminating are good for selecting adapted genotypes [58].

FPVS trials offer farmers the ability to adopt high-yielding, improved cultivars within a short time on a larger scale. Based on these FPVS trials, the cultivars BSMR 736 and BDN 711 showed constant better performance across the locations in Karnataka and Maharashtra states, respectively, due to the non-significant genotype × environment interaction (GEI) effect, so these high-yielding and stable cultivars would be mass multiplied and adopted in the respective regions. FPVS trials from other states and multi-location trials revealed significant GEI effects, indicating a differential response of genotypes in different environments. Significant genotype, environment, and GEI in pigeonpea for grain yield were reported earlier by Muniswamy et al. [59] and Arunkumar et al. [60]. For instance, FPVS trials in Andhra Pradesh, at Guntur and Prakasam, LRG 105 followed by LRG 160 while at Kurnool, LRG 52 followed by LRG 160 but across the locations, LRG 160 followed by LRG 105 revealed a significantly higher grain yield. The cultivar ICPH 2671 and ICPL 332 WR exhibited the highest grain yield at test sites and across the locations of Madhya Pradesh and Telangana, respectively.

In conclusion, advances in genetics and genomics made through TPGC are being utilized for developing new cultivars with desirable combinations of traits. The advanced backcross lines resistance to FW and SMD will be evaluated for grain yield for varietal release. The high-yielding and stably-performing genotypes in multi-location trials may be recommended for varietal release following AICRP-Pigeonpea guidelines. Similarly, high-performing and farmer-preferred varieties may be adopted in the respective districts and states by integrated efforts of different agriculture authorities, including state agricultural universities, and Kisan Vigyan Kendras.

**Supplementary Materials:** The following are available online at http://www.mdpi.com/2073-4395/10/9/1289/s1. Table S1: Details on QTLs/genomic segments identified for target traits in pigeonpea. Table S2: List of 100 elite lines used for evaluation for grain yield in pigeonpea. Table S3: Mean, range and ANOVA for grain yield of multi-location trial during cropping season 2017–2018 and 2018–2019. Table S4: Mean grain yield data of super-early duration trial during cropping season 2017–2018 and 2018–2019. Table S5: Mean grain yield data of early duration trial during cropping season 2017–2018 and 2018–2019. Table S6: Mean grain yield data of medium duration trial during cropping season 2017–2018 and 2018–2019. Table S7: Mean performance of selected cultivars of FPVS trial conducted in Karnataka for grain yield during cropping season 2017–2018 and 2018–2019. Table S8: Mean performance of selected cultivars of FPVS trials conducted in Maharashtra for grain yield during cropping season 2018–2019. Table S9: Mean performance of selected cultivars of FPVS trials conducted in Madhya Pradesh for grain yield during cropping season 2017–2018 and 2018–2019. Table S10: Mean performance of selected cultivars of FPVS trials conducted in Andhra Pradesh for grain yield during cropping season 2018–2019. Table S11: Mean performance of selected cultivars of FPVS trials conducted in Telangana State for grain yield during cropping season 2018–2019.

**Author Contributions:** Conceptualization: R.K.V. and R.K.S. along with N.S.; Methodology: A.H., C.V.S.K., A.B., A.J.H., M.S., A.R., A.M., A.N.T., C.S., S.R., D.K.P., I.P.S.; formal analysis: A.H., A.R., A.V.K.; data curation: A.H., A.R., A.V.K.; writing—original draft preparation: R.K.S., A.H., R.K.V., A.B.; writing—review and editing: R.K.S., A.H., R.K.V.; supervision: R.K.V., R.K.S., N.P.S., A.B., I.P.S.; project administration: R.K.V., N.P.S., R.K.S.; funding acquisition: R.K.V., N.P.S. All authors have read and agreed to the published version of the manuscript.

**Funding:** This research was funded by the Department of Agriculture Cooperation and Farmers Welfare, Ministry of Agriculture and Farmers Welfare, the Government of India and partly funded by Bill and Melinda Gates Foundation. This work has been undertaken as part of the CGIAR Research Program on Grain Legumes and Dryland Cereals (GLDC). ICRISAT is a member of CGIAR Consortium.

**Acknowledgments:** Authors are thankful to KB Saxena for useful discussions and suggestions on project activities and to V Suryanarayana and M Sudhakar for technical assistance in various activities.

**Conflicts of Interest:** The authors declare no conflict of interest.

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
