# Peer review of "Translational Pigeonpea Genomics Consortium for Accelerating Genetic Gains in Pigeonpea (Cajanus cajan L.)"

_agronomy, doi:10.3390/agronomy10091289_

Round 1

Reviewer 1 Report

The manuscript I received for review presents a summary of several years of scientific and breeding activities of the Translational Pigeonpea Genomics Consortium. I have only brief comments on the manuscript, especially that most of the presented results were already published by the members of the consortium. This does not diminish the value of the manuscript. In my opinion it is a significant advantage because it shows how actively the consortium works. I think that the work should be a few minor corrections.

1) A photo showing a plant, flowers, and seeds would be a good addition to the work.

2) There is no map showing the location of the sites where field experiments were conducted.

3) Line 244 is incorrectly formatted

4) There are too many horizontal lines in Figure 5a. Its readability will be improved when only the principal lines of the grid are left.

5) The description of Figure 5 is missing the space "Figure 5. Farmers..." and the same in Figure 6.

6) I have the impression that the font size is too large for the description of Figure 6. 

Best regard 

Reviewer

Author Response

Editor's comments:

It has been reviewed by experts in the field and we request that you make major revisions before it is processed further.

Authors’ Response: Authors are thankful to the Editor for arranging the Reviewer’s reports. As suggested, we have revised the MS according to the Reviewers’ suggestions and have provided our responses below in pointwise manner.

Reviewer reports:

Reviewer 1: The manuscript I received for review presents a summary of several years of scientific and breeding activities of the Translational Pigeonpea Genomics Consortium. I have only brief comments on the manuscript, especially that most of the presented results were already published by the members of the consortium. This does not diminish the value of the manuscript. In my opinion it is a significant advantage because it shows how actively the consortium works. I think that the work should be a few minor corrections.

Authors’ Response: Authors are thankful to the Reviewer for his/her positive assessment on this MS and supporting our approach as a consortium. We agree with the Reviewer that some of the trait mapping work included in this MS has been published as separate research articles (also cited appropriately in the present MS). However, this is the first time where we have synthesized all the significant achievements made through Translational Pigeonpea Genomics Consortium. We would also like to thank the Reviewer for providing valuable suggestions which helped us in improving the quality of MS.

Comment 1: A photo showing a plant, flowers, and seeds would be a good addition to the work.

Authors’ Response: We have included “Figure 5” in the revised MS. This is an overview of trials conducted with super-early pigeonpea lines at flowering stage and remaining lines at vegetative stage.

Comment 2: There is no map showing the location of the sites where field experiments were conducted.

Authors’ Response: In the revised MS, “Figure 1” has been added to show the locations used for multi-location trials.

Comment 3: Line 244 is incorrectly formatted

Authors’ Response: In the original submission file (same reviewed by Reviewer) at line number 244 “Performance and stability of genotype and environment for grain yield” is a second level heading. The similar style has been followed across the MS. For second level heading the font is normal and not bold (normal font with bold are Ist level headings) or italicized (italicized font are for IIIrd level headings).

Comment 4: There are too many horizontal lines in Figure 5a. Its readability will be improved when only the principal lines of the grid are left.

Authors’ Response: As suggested by the Reviewer, we have edited Figure and in the revised version it is Figure 7a.

Comment 5: The description of Figure 5 is missing the space "Figure 5. Farmers..." and the same in Figure 6.

Authors’ Response: In the revised version, space has been added in Figure 7 and 8 (in previous version they were Figure 5 and 6).

Comment 6: I have the impression that the font size is too large for the description of Figure 6. 

Authors’ Response: In order to make uniformity, we have changed the font size according to the main text of the MS (Palatino Linotype with font size 10).

Reviewer 2: The structure, formatting and purpose of the manuscript are a bit unclear. The authors have submitted the article as a potential "original Article", but there is no "Materials and Methods" section. In general, the manuscript is not organized with the sections required by Agronomy Instructions. The manuscript consists of a short and insufficient introduction, followed by a very large discussion section, mixed to some results. Moreover, in the affiliation section author e-mail addresses are also missing. I think the research is interesting, there are a large amount of data, however the manuscript must be reorganized, since in the present form it is not easy to follow the story.

Authors’ Response: We would like to thank Reviewer for providing encouraging remarks on this MS. This is not a Research Article, and we had discussions on this aspect with the Managing Editor. Accordingly, the structure and formatting of this MS has been followed. Nevertheless, we have provided all details to different topics in such a manner that methodology is self-explanatory. For instance, we have provided required information on methods used in analysis at appropriate places in the MS.

The objective of the article has been mentioned in the Introduction section (line number 165 - 171) as following “In summary, TPGC has been established with an aim to deploy modern genomics information for pigeonpea improvement, develop/ identify new improved varieties and to enhance the adoption of superior lines in farmers’ fields. The present article reports significant research achievements of TPGC as International Public Goods (IPGs), that will be helping and guiding future pigeonpea improvement programs. Furthermore, this article may also inspire other less-studies crop communities for taking similar consortium based approach for crop improvement.” We hope that Reviewer is happy with the Introduction.

In affiliation section, we have added email IDs of all the co-Authors. For email IDs, please see page 1, line number 10 - 35.

In view of above, we hope that the MS is acceptable for publication in the Agronomy journal.

Reviewer 2 Report

The structure, formatting and purpose of the manuscript are a bit unclear. The authors have submitted the article as a potential "original Article", but there is no "Materials and Methods" section. In general, the manuscript is not organized with the sections required by Agronomy Instructions. The manuscript consists of a short and insufficient introduction, followed by a very large discussion section, mixed to some results. Moreover, in the affiliation section author e-mail addresses are also missing.

I think the research is interesting, there are a large amount of data, however the manuscript must be reorganized, since in the present form it is not easy to follow the story.

Author Response

(The authors gave the same response as above.)

Round 2

Reviewer 2 Report

The manuscript has been improved and my comments have been properly addressed.